# Novel Putative Positive Modulators of α4β2 nAChRs Potentiate Nicotine Reward-Related Behavior

**DOI:** 10.3390/molecules26164793

**Published:** 2021-08-07

**Authors:** Skylar Y. Cooper, Austin T. Akers, Velvet Blair Journigan, Brandon J. Henderson

**Affiliations:** 1Department of Biomedical Sciences, Joan C. Edwards School of Medicine, Marshall University, Huntington, WV 25703, USA; cooper394@live.marshall.edu (S.Y.C.); akers103@live.marshall.edu (A.T.A.); journigan@marshall.edu (V.B.J.); 2Department of Pharmaceutical Sciences, School of Pharmacy, Marshall University, Huntington, WV 25701, USA

**Keywords:** allosteric, nicotine, nicotinic receptor, conditioned place preference

## Abstract

The popular tobacco and e-cigarette chemical flavorant (−)-menthol acts as a nonselective, noncompetitive antagonist of nicotinic acetylcholine receptors (nAChRs), and contributes to multiple physiological effects that exacerbates nicotine addiction-related behavior. Menthol is classically known as a TRPM8 agonist; therefore, some have postulated that TRPM8 antagonists may be potential candidates for novel nicotine cessation pharmacotherapies. Here, we examine a novel class of TRPM8 antagonists for their ability to alter nicotine reward-related behavior in a mouse model of conditioned place preference. We found that these novel ligands enhanced nicotine reward-related behavior in a mouse model of conditioned place preference. To gain an understanding of the potential mechanism, we examined these ligands on mouse α4β2 nAChRs transiently transfected into neuroblastoma-2a cells. Using calcium flux assays, we determined that these ligands act as positive modulators (PMs) on α4β2 nAChRs. Due to α4β2 nAChRs’ important role in nicotine dependence, as well as various neurological disorders including Parkinson’s disease, the identification of these ligands as α4β2 nAChR PMs is an important finding, and they may serve as novel molecular tools for future nAChR-related investigations.

## 1. Introduction

The single nucleotide polymorphisms (SNPs) of α4 and β2 receptor subunit genes (CHRNA4 and CHRNB2), which comprise a major nicotinic subtype in the brain (α4β2), are associated with heightened dependence on nicotine and initial subjective responses in both African American and youth populations [1,2]. To date, approved nicotine cessation pharmacotherapies have principally targeted α4β2 nAChRs: partial agonist, varenicline [3,4]; antagonist, bupropion [5,6]. Despite this, smoking cessation rates remain low [7], prompting the need for investigation into pharmacotherapies with a novel mechanism of action that may produce higher cessation rates. Transient receptor potential (TRP) channels have been investigated for their potential involvement in the effects of nicotine [8,9,10]. The reason for this comes from the understanding that menthol, the natural ligand for TRP melastatin 8 (TRPM8) and the most popular and widely used tobacco and e-cigarette flavor, causes several biological effects that contribute to nicotine reward and reinforcement.

One contributing effect is mediated through menthol’s interaction with TRPM8, which results in a cooling sensation that may reduce the harsh throat irritation of nicotine and tobacco [11,12,13,14,15,16]. This may contribute to smokers and vapers inhaling more nicotine, and thus may contribute to elevations in plasma nicotine concentrations [17]. In addition to its counterirritant effects, menthol has been shown to directly facilitate nicotine self-administration. Oral menthol, the TRPM8 partial agonist and cooling agent, WS-23, and cold temperatures (~11 °C) significantly increase nicotine intravenous (i.v.) self-administration in female adolescent rats, compared to nicotine alone or other tastant/odorant cues. Menthol also induces a considerable nicotine extinction burst, re-instates extinguished nicotine-seeking behavior, and acts as a conditioned cue for nicotine [16], suggesting that menthol may have direct effects on nAChRs beyond the sensory effects discussed above. In addition, constellation pharmacology efforts have identified TRPM8 and a7 nAChR co-expression in cold thermosensors from mouse and rat dorsal root ganglia and trigeminal ganglia [18]. 

In recent years, the direct effects of menthol on nAChRs have begun to be identified. Menthol enhances the nicotine-induced upregulation of nAChRs [19,20] and enhances reward-related behavior in conditioned place preference assays [19], the vapor self-administration of nicotine [21], intravenous self-administration of nicotine [16,22], and nucleus accumbens dopamine release [23]. These findings support the previous findings that menthol may be a cue-reinforcer for nicotine use [24].

Given that menthol is a well-characterized agonist of TRPM8, some have speculated that TRPM8 antagonists may be potential candidates as novel pharmacotherapies for smoking cessation, by directly affecting nicotinic pharmacology or by blocking menthol’s counter-irritant effects in relation to smoke inhalation. A novel class of menthol-derived TRPM8 antagonists has recently been discovered [25]. Based on menthol’s ability to enhance nicotine reward-related behavior, we tested these novel TRPM8 antagonists for their ability to modulate nicotine reward-related behavior using a mouse model of conditioned place preference. Here, we report that one of the most potent TRPM8 antagonists in this class (VBJ104, TRPM8 IC_50_ of 6 ± 1 nM) enhanced nicotine reward-related behavior, and this was due to its ability to act as a positive modulator (PM) of α4β2 nAChRs.

## 2. Results

We have previously shown that the TRPM8 agonist menthol enhances nicotine reward-related behavior when combined with nicotine [19], and this likely happens by directly binding to nAChRs [26]. We then decided to examine the impact of a potent TRPM8 antagonist, VBJ104 (Figure 1), on nicotine reward-related behavior in a mouse model of conditioned place preference (CPP). We hypothesized that the potent ligands of TRPM8 that displayed antagonist properties may exert the opposite effect to menthol on nicotine reward-related behavior, and result in a reduction in reward, as opposed to enhancement. We used an unbiased 10-day CPP protocol, which was identical to previously published methods [19,27,28] (Figure 2A). Mice were assigned to cohorts injected with saline, 0.5 mg/kg nicotine, 0.5 mg/kg nicotine plus 0.5 mg/kg VBJ104, or 0.5 mg/kg nicotine plus 1.0 mg/kg VBJ104. Using a one-way ANOVA, we detected a significant overall effect of drug treatment (F_(3, 36)_ = 10.1, *p* < 0.0001). 

Using a post hoc Tukey means comparison, we detected the presence of significant place preference with 0.5 mg/kg nicotine (Figure 2B). This is similar to previous reports examining nicotine reward-related behavior in mice [19,27,29]. VBJ104, a mixture of two diastereomers, is composed of >85% of the 2*SR*, 9*RS* and 10*SR* isomers (isolated, hTRPM8 IC_50_: 1.4 ± 1.0 nM). We chose a dose equivalent to that of nicotine (0.5 mg/kg VBJ104) and previous menthol investigations (1.0 mg/kg VBJ104 [19]). Here, we observed that nicotine plus 0.5 mg/kg VBJ104 produced a significant increase in reward-related behavior when compared to nicotine (*p* < 0.05; Figure 2B). Nicotine plus 1.0 mg/kg VBJ104 produced a significant CPP compared to saline (*p* < 0.0001), but not compared to nicotine alone. We observed nearly identical place preference between male and female mice for nicotine (CPP scores of 131.2 and 127.5, respectively) and nicotine plus 0.5 mg/kg VBJ104 (CPP score of 261.5 and 269.8, respectively). Given the lack of sex differences, we combined data for both males and females into Figure 2B.

Our behavioral results contradicted our hypothesis, given that we observed an enhancement in nicotine reward-related behavior. To examine how these compounds could enhance nicotine reward-related behavior, we conducted follow-up assays to determine their effects on nAChR pharmacology. To do so, we used a Ca^2+^ flux assay with neuroblastoma-2a cells transiently transfected with α4β2 nAChRs. While this cell type has been used extensively to study nAChRs in electrophysiology and microscopy assays [30,31,32], it is underutilized in fluorescence plate-reading assays compared to HEK cell lines. Therefore, we created a control nicotine concentration–response curve, and verified that our assay reproduced a nicotine EC_50_ value (81.1 ± 18.5 µM) consistent with previous literature reports, which utilized a Flexstation platform [26,33,34] (Figure 3). This EC_50_ indicates that our transient transfection of α4β2 nAChRs and our functional analysis via Ca^2+^ flux likely measure mostly low-sensitivity α4β2 nAChRs.

Next, we examined VBJ104 and five analogs (VBJ032, VBJ051, VBJ094, VBJ098, and VBJ109; see Figure 1) for their ability to alter nAChR and nicotine-induced nAChR function. As these compounds are unknowns, we used a two-addition drug application protocol (see Figure 3A), wherein the first addition included the VBJ compounds by themselves (at varying concentrations) and the second addition included the VBJ compound and 100 µM nicotine (~EC_60_). We observed no α4β2 nAChR agonist activity with any of the VBJ compounds at concentrations up to 100 µM (data not shown).

However, in combination with nicotine, we observed that the compounds enhanced nAChR function in a concentration-dependent manner (Table 1 and Figure 4). Accordingly, we classified these compounds as putative α4β2-positive modulators (PMs). VBJ104, which was most potent as an antagonist for TRPM8, showed the lowest potency as a PM for α4β2 nAChRs (EC_50_ of 4.6 µM, Table 1), but it exhibited the highest increase in efficacy (361%, Table 1). The remaining VBJ compounds exhibited much higher potencies, with a drastically reduced impact on efficacy compared to VBJ104 (Table 1 and Figure 4). VBJ098 showed no activity as an agonist, PM, or antagonist.

## 3. Discussion

Menthol acts as an agonist of TRPM8 and a NAM of α4β2 nAChRs [35]. This novel series of compounds were characterized as antagonists of TRPM8 [25]**,** and we found they failed to stimulate α4β2 nAChR activation on their own. However, they increased nAChR function in a concentration-dependent manner. Accordingly, we have deemed these compounds as putative α4β2 PMs. We acknowledge that further investigation needs to be conducted to determine if these ligands act orthosterically or allosterically. Therefore, we chose not to label these ligands as putative positive allosteric modulators (PAMs), and instead limited our designation at this time to putative PMs.

It is important to mention that there exist two types of nAChR PAMs: type-I PAMs potentiate nAChR peak-currents but have little impact on desensitization or inactivation; type-II PAMs potentiate nAChR peak-currents, and also prolong activation by enhancing slow-phase desensitization at the cost of fast-phase desensitization [36]. While we have determined that compounds such as VBJ104 can enhance agonist-induced α4β2 nAChR function, there is a need to examine the impact on desensitization and open–close channel time.

As to how VBJ104 may enhance nicotine reward-related behavior, first we can consider what is known regarding the mechanism of another TRPM8 ligand, menthol. In the case of menthol, its ability to enhance nicotine reward and reinforcement lies in its ability to alter dopamine neuron excitability [19], enhance dopamine release [23], enhance nicotine-induced upregulation of nAChRs [19,20], and act on TRPM8-related mechanisms [13]. While we have no evidence that these VBJ series compounds can alter any of these mechanisms, the ability to act as a PM on α4β2 nAChRs alone can explain how they enhance nicotine reward-related behavior. α4β2 nAChRs have been well-characterized to be critical in nicotine-related reward mechanisms [29,37,38,39]. Thus, enhancing the activity of nicotine on this subtype could have an impact on not only nicotine reward and reinforcement, but also on tolerance and sensitization.

While these ligands have no potential utility in nicotine cessation, α4β2 nAChR PMs (or PAMs) may be useful for other diseases and disorders. PAMs of nAChRs have been implicated for their potential use in treatment of schizophrenia [40] and cognitive disabilities [41]. Given that menthol exerts an effect on all subtypes of nAChRs and many members of the Cys-loop superfamily [35,42,43,44,45,46], follow-up studies for these VBJ compounds’ activity on other nAChR subtypes and ligand-gated ion channels may be necessary.

While the results of this study did not follow our original hypothesis, we have discovered a new series of α4β2 nAChR PMs that may be useful as novel probes. As discussed above, examinations of activity on the major nAChR subtypes, and possibly other members of the Cys-loop superfamily, must be carried out. Additionally, another potential follow-up for this work would be to expand our concentration–response studies to determine if VBJ104 may have a concentration-dependent dual effect (see Figure 4B_1_). Similarly, there needs to be an expanded dose range for our CPP assays. Currently, the higher dose of VBJ104 (1.0 mg/kg) produces a lesser response than the 0.5 mg/kg dose. Nicotine exhibits an inverted-U dose response in CPP assays, exhibiting a peak of reward-related behavior followed by aversion-related behavior at higher doses. Thus, higher doses of VBJ104 may potentiate nAChR actions to a degree that produces a similar aversion-related response. Thus, while we have failed to identify a novel chemical scaffold for nicotine cessation, we may have discovered compounds that are useful in other areas of interest. This will require careful examination via assays related to learning, memory, and anxiety-related behaviors.

## 4. Materials and Methods

### 4.1. Mice

All experiments were conducted in accordance with the Guide for Care and Use of Laboratory Animals provided by the National Institutes of Health. Protocols were approved by the Institutional Animal Care and Use Committee at Marshall University. Adult male and female wildtype C57BL/6J mice (3–5 months old) were obtained from the Jackson Laboratory (https://www.jax.org/strain/000664, accessed on 7 November 2019). Mice were kept on a standard 12/12 h light/dark cycle at 22 °C and given food and water ad libitum.

### 4.2. Reagents and Dose Selection

The calcium-sensitive fluorescent probe, Calcium 6, was obtained from Molecular Devices (Sunnyvale, CA, USA). Minimum essential medium (MEM) was obtained from Corning. Opti-MEM, penicillin and streptomycin were obtained from Invitrogen Corporation (Grand Island, NY, USA). Nicotine ditartrate dihydrate (product # 415660500) was obtained from Acros Organics (Fair Lawn, NJ, USA). We utilized a nicotine dose of 0.5 mg/kg (with respect to free base) for its previously determined rewarding effect for mice in conditioned place preference assays [19,29]. VBJ series compounds (see Table 1) were prepared as described previously [25]. All molecules were >99.6% pure, as determined by elemental analysis. For pharmacological evaluation, all compounds were initially dissolved in 100% DMSO (0.01 M stocks) due to solubility. Further dilutions of compounds were made in double-distilled H_2_O or extracellular solution (ECS) (≤100 μM).

### 4.3. Conditioned Place Preference (CPP) Assays

CPP assays were completed in a three-chamber spatial place preference chamber (Harvard Apparatus, PanLab, dimensions: 47.5 × 27.5 × 47.5 cm) using male and female C57BL/6J mice. Time in chambers was recorded by motion tracking software (SMART 3.0). A 10-day, unbiased protocol identical to previous studies [19,28] was used where drugs (saline, nicotine (0.5 mg/kg), nicotine plus 0.5 mg/kg VBJ104, and nicotine plus 1.0 mg/kg VBJ104) were given immediately before confinement in the right white/grey chamber on drug days, and saline was given immediately before confinement in the left white/black chamber on saline days (via intraperitoneal injections). On day 1, a pre-test was completed wherein mice were placed in the central chamber and allowed free access to the apparatus for 20 min. Mice that spent >65% of the test in one chamber were excluded and the remaining mice were counterbalanced. For counterbalancing, mice were separated into groups of approximately equal bias, similar to previously published CPP methods [47]. No exclusions were necessary for these studies. Following counterbalancing, no initial biases were noted. The mice received their designated drug injections on days 2, 4, 6, and 8, and received saline injections on days 3, 5, 7, and 9. Each conditioning period lasted 20 min. On day 10, a post-test was completed whereby the mice were again placed in the central chamber and allowed free access for 20 min. In total, 5 male and 5 female C57BL/6J mice, 3–5 months old, were used in the CPP assays for each treatment group. Time spent in in the saline-paired chamber was subtracted from time spent in the drug-paired chamber to score the pre-test and post-test. CPP score (or change from baseline) was determined by subtracting the pre-test score from the post-test score. A significant positive CPP score is indicative of reward-related behavior, while a significant reduction is indicative of aversion-related behavior.

No sex differences were observed and data for males and females were combined (see *Results* for specifics). Data are expressed as a change in baseline preference, which was analyzed using a one-way ANOVA with a post hoc Tukey.

### 4.4. Neuro-2a Cell Culture and Transient Transfections

Mouse neuroblastoma-2a (neuro-2a) cells were cultured using standard techniques. Cells were maintained in minimum essential medium (MEM, product #10-010-CV obtained from Corning) plus 10% fetal bovine serum, 100 IU/mL penicillin, and 100 µg/mL streptomycin at 37 °C and 5% CO_2_ in a humidified incubator. Cells were plated at a density of 1.5–2.0 × 10^5^ cells per well in clear 96-well culture plates previously coated with poly-l-ornithine. At 24 h after plating, neuro-2a cells were transfected with α4 and β2 nAChR subunits using Lipofectamine 3000 (Invitrogen) following manufacturer recommendations in Opti-MEM. The plasmid concentrations used for transfection were 5 µg of α4 and β2 (mouse) nAChR subunits for each 96-well plate. At 24 h after transfection, the 96-well plates were washed and replaced with standard culture medium. At 24 h after replacing with standard culture medium, 96-well plates were used in Flexstation assays.

### 4.5. Calcium 6 Assay (Calcium Accumulation Assay)

The Calcium 6 procedure was carried out via a previously published procedure with minor modifications using calcium 5 [48,49,50]. For this calcium accumulation assay, neuro-2a cells transiently expressing mouse α4β2 nAChRs were used (see above for transfection methods). On the day of the experiment, cells were incubated in the dark for 2 h at 24 °C with 50% Calcium 6 NW dye (Molecular Devices). The plates were then placed into a fluid handling integrated fluorescence plate reader (Flexstation III, Molecular Devices, Sunnyvale, CA, USA) and fluorescence was read at an excitation of 485 nm and emission of 525 nm from the bottom of the plate with changes in fluorescence monitored at ~0.8 s intervals. Baseline fluorescence was monitored for 20 s and then two drug additions (first at 20 s and the second at 60 s) were applied using a Flexstation application speed of 2. At the beginning of the Flexstation assays, each well in the 96-well plate started with 100 µL of solution. For nicotine concentration response, the first addition contained only assay buffer (50 µL), and the second addition contained nicotine (50 µL) at 4× the target concentration.

For assays examining the VBJ compounds, potential PM activity was assessed using the following protocol. For the nicotine control group, assay buffer (50 µL) was added in the first addition, and nicotine (50 µL of a 400 µM solution) was added to achieve a final concentration of 100 µM. Treatment groups received the VBJ compound (50 µL of a 3× solution) in the first addition and then the same nicotine solution (400 µM) with the desired concentration of the VBJ compound (1×) in the second addition. Sham-treated groups were only given assay buffer.

### 4.6. Calculations

Functional responses were quantified by first calculating the net fluorescence (the difference between control sham-treated and control agonist-treated groups). Results were expressed as a percentage of control (100 μM nicotine). For each PM, six concentrations were used in a series of concentration–response studies. Following transformation to log values, sigmoidal-varied slope curves were fit to data using Prism 9 with no constraints (Graphpad, San Diego, CA, USA). From these curves, EC_50_ and maximal changes in efficacy were determined for each PM. Functional data were calculated from the number of observations (*n*) performed in triplicate. Due to the use of log values in calculating the EC_50_ values, geometric (as opposed to arithmetic) means were calculated for PMs in this study. All EC_50_ values are expressed as geometric means (95% confidence limits). Due to solubility problems, compound concentrations greater than 100 μM were not used in our concentration–response studies with VBJ compounds. The DMSO concentration at this compound concentration was ≤1%, and this had no effects on basal- or agonist-induced increases in fluorescence intensity.

### 4.7. Statistical Analysis

All results are presented as mean ± SEM and all statistical analyses were performed using GraphPad Prism 9. Data were analyzed using a one-way ANOVA. When effects were shown to be significant, a post hoc Tukey test was performed to compare the individual drug treatment groups. For CPP assays, males and females were analyzed separately using a two-way ANOVA. No sex differences were noted; therefore, sexes were combined.

### 4.8. Supplemental Methods

Supplemental methods and data are available in the Appendix A.

## Figures and Tables

**Figure 1 molecules-26-04793-f001:**
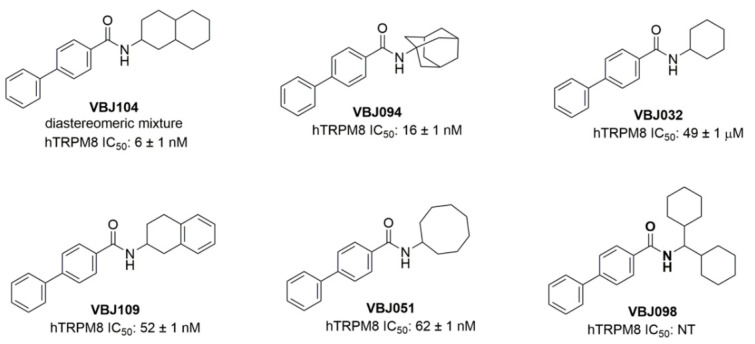
Structures of novel VBJ series compounds. The IC_50_ values of hTRPM8 are from a previously published report [25].

**Figure 2 molecules-26-04793-f002:**
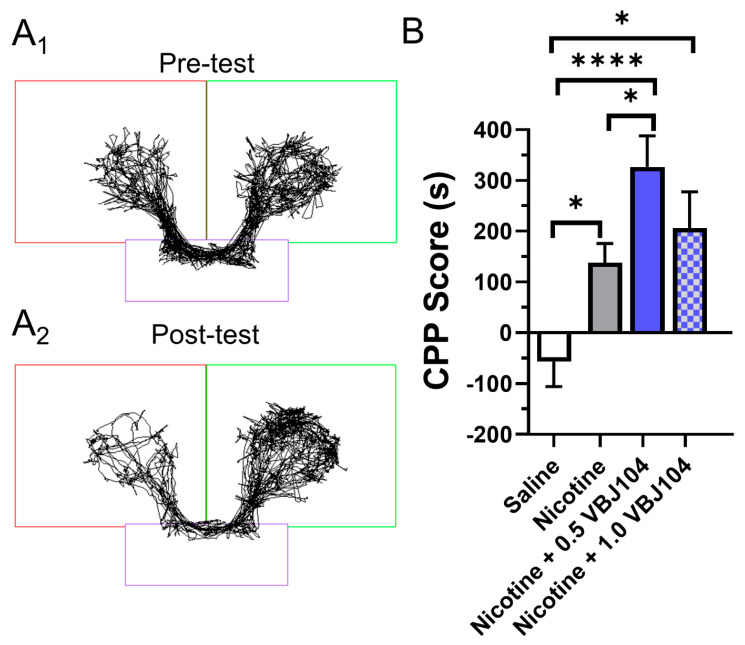
(**A_1_**,**A_2_**) Representative time-traces for a mouse used in pre- and post-tests in a conditioned place preference assay and assigned to the 0.5 mg/kg nicotine treatment group. (**B**) Male and female mice were assigned saline, 0.5 mg/kg nicotine, 0.5 mg/kg VBJ104 plus 0.5 mg/kg nicotine, or 1.0 mg/kg VBJ104 plus 0.5 mg/kg nicotine, and used in a CPP assay (via intraperitoneal injections; *n* = 10 mice per condition, 5 males and 5 females). * *p* < 0.05; **** *p* <0.0001.

**Figure 3 molecules-26-04793-f003:**
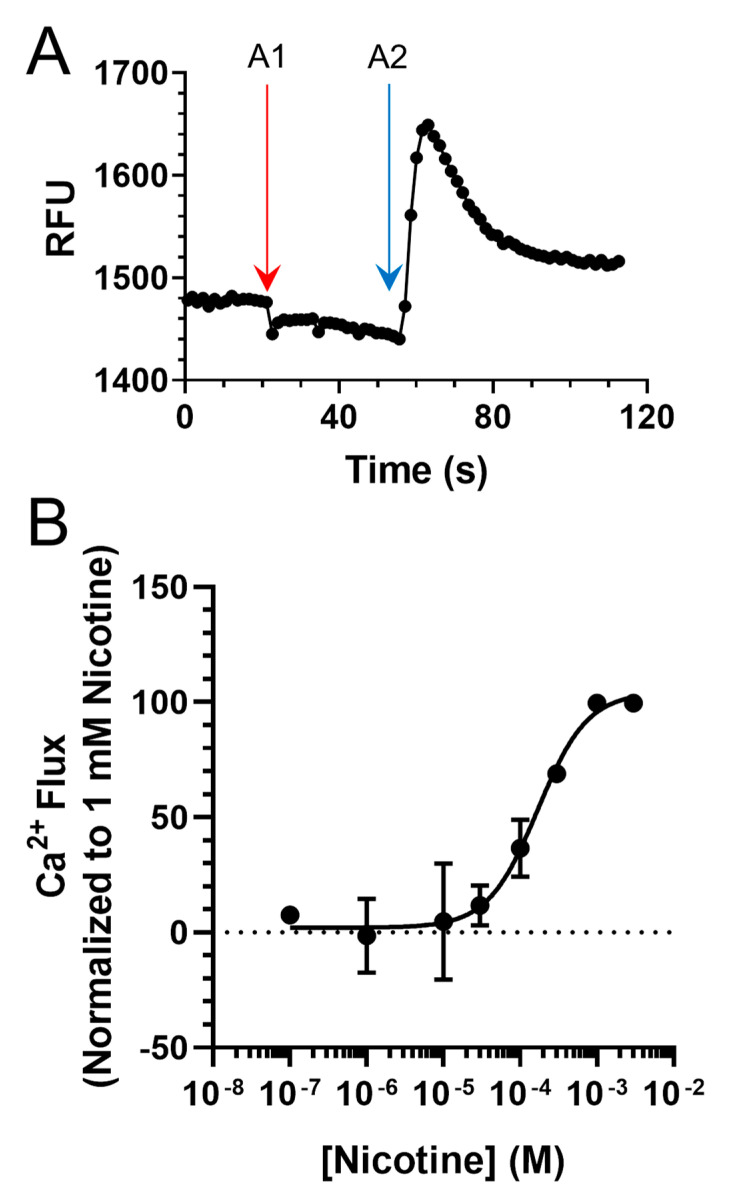
(**A**) Representative Ca^2+^ Flux trace from a single 96-well plate seeded with cells transiently transfected with mouse α4β2 nAChRs. A1 and A2 designate drug additions 1 (vehicle) and 2 (100 µM nicotine). (**B**) Concentration–response of nicotine on neuroblastoma-2a cells transiently transfected with α4β2 nAChRs. Data are mean ± SEM and are normalized to 1 mM nicotine. For B, *n* = 6 individual experiments.

**Figure 4 molecules-26-04793-f004:**
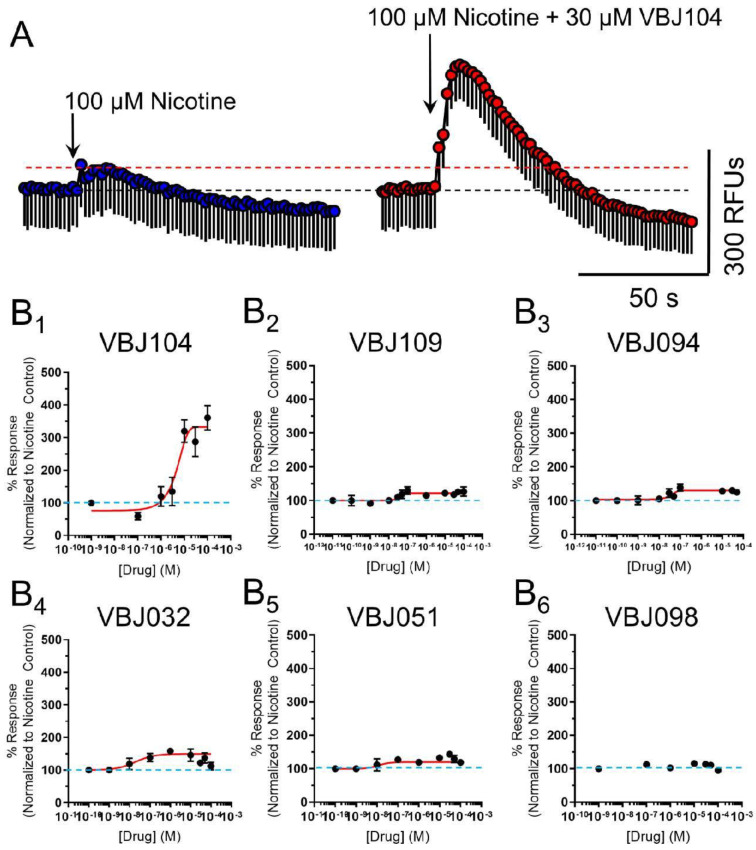
(**A**) Representative Ca^2+^ flux from a 100 µM application of nicotine (left) and a 2 µM application of nicotine in the presence of 30 µM VBJ104 on cells transiently transfected with mouse α4β2 nAChRs. Data are mean ± SEM (triplicate data points). (**B**_1_–**B**_6_) Concentration–response curves for varying concentrations of VBJ ligands in the presence of 100 µM nicotine. For **B**_1_–**B**_6_, *n* = 4–7 individual experiments.

**Table 1 molecules-26-04793-t001:** In vitro Ca2^+^ flux data.

Compound	hTRPM8 IC_50_ *^a^*	α4β2 nAChR PM EC_50_ *^b^*	Max Efficacy (Normalized to 100 µM Nicotine) *^c^*
VBJ032	49 ± 1 µM	77.6 nM	158.5 ± 8.6% (50 µM)
VBJ051	52 ± 1 nM	8.2 (0.8–47.2) nM	143.6 ± 7.1% (50 µM)
VBJ094	16 ± 1 nM	23.6 (1.8–68.2) nM	130.7 ± 7.8% (30 µM)
VBJ098	NE	NE	NE
VBJ104	6 ± 1 nM	4.6 (2.6–8.4) µM	360.7 ± 37.7% (100 µM)
VBJ109	52 ± 1 nM	29.7 (4.3–69.0) nM	125.6 ± 5.1% (50 µM)

*^a^*, the hTRPM8 IC_50_ data are derived from [25]. *^b^*, data are expressed as means with 95% confidence limits. *^c^*, the concentration at which maximum increase occurs is indicated in parenthesis. NE, no detectable effect. *n* = 4–7 individual experiments for each compound.

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
