# Peer review of "Novel Putative Positive Modulators of α4β2 nAChRs Potentiate Nicotine Reward-Related Behavior"

_molecules, 2021, doi:10.3390/molecules26164793_

Round 1

Reviewer 1 Report

The MS reads well. The design and results were clear and easy to understand. I have a few minor comments.

line 34: I would add TRP before melastatin 8 to make the name complete (e.g., TRP melastatin 8 (TRPMS)

line 43: I would define WT-23 in some way (e.g., the cooling agent WS-23)

Figure 2. The legend should state the p values for each asterisk even though p values are  stated in the text

Figure 4 A: VBJ104, not 1104

Author Response

line 34: I would add TRP before melastatin 8 to make the name complete (e.g., TRP melastatin 8 (TRPMS)

Response: The above correction has been made

line 43: I would define WT-23 in some way (e.g., the cooling agent WS-23)

Response: The above correction has been made

Figure 2. The legend should state the p values for each asterisk even though p values are  stated in the text

Response: The above correction has been made

Figure 4 A: VBJ104, not 1104

Response: The above correction has been made

Reviewer 2 Report

The present work, performed by Skylar Y. Cooper, Austin T. Akers, V. Blair Journigan, and Brandon J. Henderson, studies new compounds characterized as antagonists of the melastatin 8 transient receptor potential (TRPM8) channel.

They were tested on both nicotine reward-related behavior and the functioning of mouse α4β2 nicotinic acetylcholine receptors (nAChRs).

The main findings were that these ligands: a) enhanced nicotine reward-related behavior, opposite to the expected result, and b) increased calcium flux exerted by nicotine on α4β2 nAChRs, possibly acting as positive modulators.

The subject is interesting and contributes for understanding the effects and mechanisms of action of these new compounds on α4β2 nAChRs.

Comments for the manuscript:

Title and Abstract: As the authors comment at the beginning of Discussion (lines 146-148); there are no evident data indicating that these compounds act as allosteric modulators on α4β2 nAChRs, the Title could be change to “Novel positive modulators of α4β2 nAChRs potentiate nicotine reward-related behavior”.

The last sentence of the Abstract is very general and do not give information. It would be beneficial to give some conclusion regarding the results.

Introduction:

The second sentence of the Introduction (lines 28, 29) regarding the α4β2 partial agonist varenicline used for nicotine cessation treatment, would benefit with one reference. At this respect, take in consideration that the antidepressant bupropion that acts as antagonist for a variety of nAChRs (opposite effect of varenicline) is also used for nicotine cessation treatment (Slemmer et al. Bupropion is a nicotinic antagonist. J Pharmacol Exp Ther. 2000; 295: 321-327).

Lines 32-34: “A variety of agonists and antagonists for TRPM8 have been explored in nicotine addiction-related behaviors [6-10]”. At first glance, from the titles of these references, these studies are related only with menthol.

Additionally, and interestingly, there is a recent article that identifies a variety of target molecules for menthol, including channels and receptors (Umezu. Identification of novel target molecules of l-menthol. Heliyon. 2021 Jun 17;7(6): e07329). Part of this information could be incorporated in the introduction and discussion.

It would be beneficial to briefly mention properties of the positive allosteric modulators (PAMs) types I and II.

Results:

Lines 107-110: “…our assay reproduced a nicotine EC50 value (81.1 ± 18.5 μM) that is consistent with previous literature reports that utilized a Flexstation platform [28, 29]”. Data of nicotine EC50 value in this study is very different from those reported in references 28 and 29. It is comparable with that in reference 21, for low-sensitivity α4β2 nAChRs: ACh EC50 was 93.4 μM.

Fig. 4B: Replace VBJ1104 for VBJ104.

Fig. 4B1: The nicotine response in the presence of 100 nM VBJ104 is much less than control nicotine response; it seems to be an inhibitory effect of VBJ104. If possible, please discuss this finding in the context of a possible dual effect of this substance.

Fig. 4B2-6: This reviewer suggests to graphing the ordinate axis from 0 to 200%, for better visualization of the results.

For future works, one possibility for electrophysiological testing these new compounds could be the oocyte preparation, as in reference 21.

Discussion:

Regarding results from Fig. 2B (lines 88-91), how is it explained that a higher dose of VBJ104 (1.0 mg/kg) produced less effect on conditioned place preference assay than 0.5 mg/kg VBJ104. Does it have been observed for other PAMs of nAChRs?

Third paragraph (lines 158-169): It seems that this paragraph, together with supplementary material are not in the context of the results. One alternative is to give arguments to put them in context, mainly for behavioral results.

It may be beneficial to discuss previous findings regarding PAMs of α4β2 nAChRs with the results obtained here.

Last paragraph (lines 177-183): Very general information. Possibly it is better to end with some conclusions together with the mentioned perspectives.

Author Response

Title and Abstract: As the authors comment at the beginning of Discussion (lines 146-148); there are no evident data indicating that these compounds act as allosteric modulators on α4β2 nAChRs, the Title could be change to “Novel positive modulators of α4β2 nAChRs potentiate nicotine reward-related behavior”.

 Response: We acknowledge this fact highlighted by the reviewers and agree that we do not provide data supporting allosteric actions. Accordingly we have changed the title as suggested by the reviewer.

The last sentence of the Abstract is very general and do not give information. It would be beneficial to give some conclusion regarding the results.

Response: The Reviewer is correct. In the interest of removing vague statements, we have elected to modify the abstract to include our previous statements only up to the point were we summarize our research findings.

Introduction:

The second sentence of the Introduction (lines 28, 29) regarding the α4β2 partial agonist varenicline used for nicotine cessation treatment, would benefit with one reference. At this respect, take in consideration that the antidepressant bupropion that acts as antagonist for a variety of nAChRs (opposite effect of varenicline) is also used for nicotine cessation treatment (Slemmer et al. Bupropion is a nicotinic antagonist. J Pharmacol Exp Ther. 2000; 295: 321-327).

 Response: This section of the introduction has been modified to increase clarity and provide further context for both varenicline and bupropion as partial agonists and antagonists, respectively.

Lines 32-34: “A variety of agonists and antagonists for TRPM8 have been explored in nicotine addiction-related behaviors [6-10]”. At first glance, from the titles of these references, these studies are related only with menthol.

Response: The reviewer is correct and this statement in the introduction has been removed.

It would be beneficial to briefly mention properties of the positive allosteric modulators (PAMs) types I and II.

Response: A brief mention of type-I and type-II PAMs has been added to the discussion. 

Results:

Lines 107-110: “…our assay reproduced a nicotine EC50 value (81.1 ± 18.5 μM) that is consistent with previous literature reports that utilized a Flexstation platform [28, 29]”. Data of nicotine EC50 value in this study is very different from those reported in references 28 and 29. It is comparable with that in reference 21, for low-sensitivity α4β2 nAChRs: ACh EC50 was 93.4 μM.

 Response: Yes it is and we have no included this reference to the above statement in question.

Fig. 4B: Replace VBJ1104 for VBJ104.

 Response: The above Correction has been made

Fig. 4B1: The nicotine response in the presence of 100 nM VBJ104 is much less than control nicotine response; it seems to be an inhibitory effect of VBJ104. If possible, please discuss this finding in the context of a possible dual effect of this substance.

 Response: We have included in our revised discussion a new statement regarding the potential dual effect of this compound.

Fig. 4B2-6: This reviewer suggests to graphing the ordinate axis from 0 to 200%, for better visualization of the results.

 Response: We acknowledge that the scales matched to VBJ104 may make the data for the other ligands compressed; but several internal/external reviewers of our work have convinced us that keeping the y-axis matched for all the ligands is critical. Therefore, we have elected to keep the axis as displayed.

For future works, one possibility for electrophysiological testing these new compounds could be the oocyte preparation, as in reference 21.

Response: that is an excellent suggestion in which we are considering for follow-up studies. 

Discussion:

Regarding results from Fig. 2B (lines 88-91), how is it explained that a higher dose of VBJ104 (1.0 mg/kg) produced less effect on conditioned place preference assay than 0.5 mg/kg VBJ104. Does it have been observed for other PAMs of nAChRs?

 Response: This is now mentioned in the revised Discussion

Third paragraph (lines 158-169): It seems that this paragraph, together with supplementary material are not in the context of the results. One alternative is to give arguments to put them in context, mainly for behavioral results.

 Response: Given that the data referenced here is all presented in the supplementary material, we have elected to move this discussion to the supplement as well.